# Plant Defense and Viral Counter-Defense during Plant–Geminivirus Interactions

**DOI:** 10.3390/v15020510

**Published:** 2023-02-12

**Authors:** Jianhang Zhang, Mengyuan Ma, Yule Liu, Asigul Ismayil

**Affiliations:** 1Key Laboratory of Xinjiang Phytomedicine Resource and Utilization of Ministry of Education, College of Life Sciences, Shihezi University, Shihezi 832003, China; 2MOE Key Laboratory of Bioinformatics, Center for Plant Biology, School of Life Sciences, Tsinghua University, Beijing 100084, China; 3Tsinghua-Peking Joint Center for Life Sciences, Tsinghua University, Beijing 100084, China

**Keywords:** geminiviruses, gene silencing, defense, counter-defense, autophagy

## Abstract

Geminiviruses are the largest family of plant viruses that cause severe diseases and devastating yield losses of economically important crops worldwide. In response to geminivirus infection, plants have evolved ingenious defense mechanisms to diminish or eliminate invading viral pathogens. However, increasing evidence shows that geminiviruses can interfere with plant defense response and create a suitable cell environment by hijacking host plant machinery to achieve successful infections. In this review, we discuss recent findings about plant defense and viral counter-defense during plant–geminivirus interactions.

## 1. Introduction

Plants pathogens, including viruses, are responsible for many diseases and cause significant losses of agricultural production [1,2,3]. *Geminiviridae* is one of the largest and most important families of plant viruses with small circular, single-stranded DNA that are 2.7–5.2 kb in size. These viruses infect a wide range of plant species and are a major threat to almost all economically important crops and food security. Viruses of the family *Geminiviridae* are divided into 14 genera based on their genome organization, host range, and insect vectors (ictv.global/report/geminiviridae). Currently, the family *Geminiviridae* includes more than 500 species. The genome of geminivirus can be either monopartite (a single DNA component) or bipartite (two DNA components: DNA A and DNA B). For effective infection, geminivirus encodes 6-8 multifunctional proteins, which are required for viral replication, the assembly of virus particles, cell-to-cell movement, and viral symptom induction. The replication initiator protein (Rep) encoded by ORF AC1/C1 (also called AL1/L1) is essential in virus rolling-circle replication, stimulates virus transcription and suppresses host gene silencing (transcriptional gene silencing). The geminiviral transcriptional activator protein (TrAP) encoded by ORF AC2/C2 acts as a silencing suppressor (both transcriptional gene silencing (TGS) and post-transcriptional gene silencing (PTGS)) and involved in symptom development, suppression of HR and inhibition of hormone-mediated defense. ORF AC3/C3 encodes a replication enhancer protein (REn), which interacts with Rep and enhances viral DNA accumulation and symptom development. AC4/C4 ORF contained entirely within the AC1 ORF, but in a different frame, encodes a multifunctional protein called AC4/C4. Geminiviral AC4/C4 proteins are critical in the suppression of gene silencing (both TGS and PTGS) and HR, regulation of cell cycle and cell division, symptom development and viral systemic movement. Coat protein (CP) is encoded by ORF AV1/V1, is a structural protein to geminiviral particles and it has been associated with virus genome packaging, insect transmission and the cell-to-cell and systemic spread of viruses. It also serves as a nucleocytoplasmic shuttling protein in monopartite viruses. The AV2/V2 protein is a pathogenicity determination factor, a silencing suppressor (both TGS and PTGS) and a movement protein of geminiviruses. The DNA-B component contains two genes, BC1 and BV1, that encode two proteins, MP and NSP, respectively, which are involved in the intercellular and intracellular movement of viral particles. Geminiviruses are often associated with additional small circular single-stranded DNA molecule referred to as satellites. Satellites are approximately half the size of geminivirus DNA genomes. Alpha and deltasatellites are associated with both monopartite and bipartite begomoviruses, whereas betasatellites are associated with monopartite begomoviruses only. Alphasatellites encode replication initiator proteins and have not been shown to play a crucial role in symptom development or pathogenicity. Betasatellites are pathogenicity determinants and depend completely on their helper virus for replication and encapsidation. The only protein encoded by betasatellites is βC1, which is essential in pathogenicity determination, silencing suppression (both transcriptional gene silencing and post-transcriptional gene silencing), systemic movement and suppressing host defense. Deltasatellites do not encode any proteins but some of them affect viral DNA accumulation and symptomatology (genus: Begomovirus, ICTV). Additional small proteins AC5/C5 or V3 from geminiviruses are identified as symptom inducers and silencing suppressors. These proteins also reprogram plant cell cycle and transcriptional control, inhibit cell death pathways, interfere with cell signaling and protein turnover and suppress plant defense. 

In the course of co-evolution, plants have evolved multilayered antiviral immune systems, including RNA silencing, pathogen-associated molecular pattern (PAMP)-triggered immunity (PTI) and effector-triggered immunity (ETI). RNA silencing triggered by geminivirus infection can target either viral RNAs for degradation at the post-transcriptional level or viral DNAs for epigenetic modification at the transcriptional level to inhibit viral replication and pathogenicity [4]. To counter plant defense, geminiviruses can encode different viral proteins, such as AC1/C1, AC2/C2, AC4/C4, V2, AC5/C5 and βC1, to inhibit various steps in post-transcriptional gene silencing and transcriptional gene silencing pathways [5,6,7,8,9]. Beside RNA silencing, plants also develop protein-kinase-mediated antiviral immunity, effector-triggered immunity, autophagy-mediated antiviral defense, a ubiquitin-proteasomal protein-degradation system and hormone-mediated defense to defeat geminivirus [10]. However, geminiviruses can also evade or subvert these plant defense mechanisms for their own benefits. 

## 2. Antiviral RNA Silencing

RNA interference (RNAi) is a well-established, conserved gene silencing process mediated by small RNAs among plants, animal and fungi. It is a common defense mechanism against invasive nucleic acids, such as transposons, transgenes and viral genome, or its transcripts [11,12,13]. In this process, plants launch defenses against viruses by targeting viral RNA for degradation or translation inhibition through PTGS, or epigenetic modification, including DNA methylation or histone modification through TGS [14] (Figure 1). Geminiviruses depend on the host system to replicate through double-stranded DNA intermediates and associate with cellular histone proteins to form its minichromosomes [15]. During geminivirus infection, DNA methylation-mediated TGS is induced and targets geminiviral genome DNA [16,17]. The genomic DNA of many geminiviruses, such as *Beet curly top virus* (BCTV), *Beet severe curly top virus* (BSCTV), *Cabbage leaf curl virus* (CaLCuV), *Tomato yellow leaf curl China virus* (TYLCCNV), *Tomato leaf curl Yunnan virus* (TLCYnV) and *Cotton leaf curl Multan virus* (CLCuMuV), are methylated during viral infection [15,18,19,20,21]. Plants utilize TGS to suppress viral minichromosomes and silence viral gene expression by RNA-directed DNA methylation (RdDM) [22]. The RdDM pathway includes several steps, as follows: double-stranded RNAs (dsRNAs) produced by RNA Polymerase IV (Pol IV) and RNA-dependent RNA polymerase 2 (RDR2) are spliced by DCL3 to generate 24 nt small interfering RNAs (siRNAs). These siRNAs are then stabilized by HUAENHANCER 1 (HEN1) and loaded into Argonaute 4 to form AGO4/siRNAs complexes. These complexes further recruit domain-rearranged methyl transferase 2 (DRM2) to methylate the target viral genomic DNA [23,24,25,26]. Several studies demonstrate the significance of the RdDM pathway in anti-geminiviral immunity [4,15,16,17,18,19,20,21,22,23,24,25,26]. Distinct geminiviral genome methylation is reduced in *Arabidopsis thaliana* mutants ddm1, ago4, drm1, drm2, cmt3, adk1 and dcl3 10. In addition, DNA methylation has also been found to be associated with symptomatic recovery caused by geminivirus infection [15,27]. AGO4 binds directly to virus-derived siRNAs (vsiRNAs) and mediates the methylation of viral DNA to attenuate geminivirus infection [9,12]. Interestingly, a typical dominant resistance gene Ty-1 is shown to enhance the transcriptional gene silencing of geminiviruses. Ty-1 encodes γ class RNA-dependent RNA polymerase (RDR) and promotes plants to produce more viral small interfering RNAs (vsiRNAs) complementary to the virus genome, leading to a higher amount of cytosine methylation of viral genomic DNAs, enhanced TGS and stronger plant resistance to *Tomato yellow leaf curl virus* (TYLCV) and other geminiviruses [28]. 

In addition, histone modification also plays a crucial role in plant defense against geminivirus infection. H3K9 histone methyltransferase KRYPTONITE (KYP) controls viral chromatin methylation and maintains TGS to combat virus, and the repression of KYP enhances virus survival in the host [29,30]. *Arabidopsis* histone reader EML1 (EMSY-LIKE 1) represses viral gene expression and virus infection by inhibiting the association of RNA polymerase II with viral chromatin [31]. 

Plants deploy PTGS as another layer of defense against RNA viruses and DNA viruses, whereas TGS targets virus DNA. Plant PTGS pathways include the following: (1) the formation of double-stranded RNA (dsRNA) from internal base-paired stem-loop RNA structures, form transposons, transgenes or RNA-dependent RNA polymerase (RDRs)-directed synthesis from single-stranded RNA (ssRNA); (2) the cleavage of dsRNAs into small interfering RNAs (siRNAs) by Dicer family proteins; (3) siRNAs are loaded into an RNA-induced silencing complex (RISC); and finally, (4) the sequence-specific degradation of target mRNAs and the inhibition of transcription [4,32]. In the cytoplasm, RISC mediates PTGS to inhibit the transcription of viral genes via the degradation of viral mRNAs. PTGS is induced during geminivirus infection, and some geminiviruses are developed as virus-induced gene silencing (VIGS) vectors [33,34,35,36,37]. The expression of multiple PTGS-related genes is upregulated during viral infection [38]. Upon virus infection, plants can sense calcium flux triggered by virus intrusion to promote the interaction between calmodulin (CaM) and CaM-binding transcription activator 3 (*CAMTA3), inducing the* expression of RDR6 and BN2. BN2 can degrade some miRNAs to stabilize levels of AGO1/2 and DCL1 mRNA to promote PTGS [38]. Suppressor of gene silencing 3 (SGS3) is a plant-specific RNA-binding protein that cooperates with RDR6 to trigger geminivirus-induced gene silencing and suppress several geminivirus infections [39]. CaLCuV is targeted by subsets of DCLs. DNA virus-derived small interfering RNAs (siRNAs) of specific size classes (21, 22 and 24 nt) are produced by all four DCLs, including DCL1, known to process microRNA precursors [40]. In PTGS-mediated antiviral defense, DCL2 and DCL4 usually process dsRNA precursors into 21 and 22 nt siRNAs, and then these siRNAs interact with AGO1 and AGO2. PTGS mainly cleaves viral RNAs through the nucleic acid endonuclease activity of AGO1 for antiviral purpose [34].

## 3. Geminiviral Suppressors of Gene Silencing

RNA silencing is a general antiviral defense mechanism against viruses, including geminiviruses. However, geminiviruses have evolved counter-defense mechanisms to overcome plant RNA silencing by encoding viral suppressors of RNA silencing (VSRs). Many geminivirus-encoded proteins are capable of suppressing the PTGS and TGS pathway [41,42,43,44,45] (Figure 1). Rep, which is also designated AC1/C1, from different geminiviruses suppresses TGS by reducing the expression of plant DNA methyltransferases [46]. Geminiviral AC2/AL2 proteins interact with and inactivate different silencing factors, such as adenosine kinases (ADKs), H3K9me2 histone methyltransferase and SU(VAR)3-9homolog 4/kryptonite (SUVH4/KYP), to diminish plant TGS [30,47,48]. BSCTV C2 attenuates the degradation of S-adenosyl-methionine decarboxylase 1 (SAMDC1), a key enzyme for the synthesis of polyamines in mammals and plants, to suppress DNA methylation-mediated gene silencing [20]. CLCuMuV C4 suppresses both transcriptional and post-transcriptional gene silencing by interacting with and inhibiting SAM synthetase enzyme activity [49]. C4 also interacts with AGO4 and eliminates viral genome methylation [50]. AC5 from *Mungbean yellow mosaic India virus* (MYMIV) interferes with TGS by reducing DNA methylation through the repressing expression of a CHH cytosine methyltransferase [7]. CLCuMuV V2 counters RdDM-mediated TGS antiviral defense by directly interacting with AGO4 to facilitate virus infection [9]. TYLCV V2 interacts with host histone deacetylase 6 and interferes with the recruitment of MET1 to decreases viral genome methylation [51]. The TYLCCNV βC1 protein also represses cytosine methylation by interacting with S-adenosyl homocysteine hydrolase (SAHH), a methyl cycle enzyme required for SAM production and methylation-mediated TGS [19]. The TYLCCNB βC1 protein also interacts with ROS1-like DNA glycosylase and with DEMETER (DME) DNA glycosylase, while facilitating DNA glycosylase activity to decrease viral DNA methylation and promote viral virulence [52]. These studies suggest that geminiviruses may disturb the proper functions of the cellular methyl cycle and affect TGS.

Geminiviruses also encode VSRs to inhibit plant antiviral PTGS defense. The Mastrevirus-encoded Rep protein binds to 21 nt single-stranded and double-stranded viral siRNAs to inhibit host PTGS [45]. AC2 proteins encoded by different geminiviruses can interact with AGO1, RDR6 and the calmodulin-like protein (rgs-CaM), an endogenous suppressor of PTGS, to suppress RNA silencing [53,54,55]. Geminiviral C4 protein inhibits the intercellular spread of 21 nt viral siRNA for interfering with host RNA silencing [56]. V2 proteins from geminiviruses suppress PTGS while inhibiting the suppressor of gene silencing 3 (SGS3) and impairing the RDR6/SGS3 pathway [57,58]. TYLCCNV V2 disrupts siRNAs generated against the virus and hinders the silencing pathway [59]. Additionally, CLCuMuV V2 sequesters long dsRNA and prevents its Dicer-mediated cleavage, and V2 can also disrupt calmodulin-CAMTA3 interaction to counteract PTGS defense [38,60]. Transgenic plants infected with TYLCV or cotton leaf curl Multan betasatellite (CLCuMuB). βC1 expression shows an increased level of AGO1 and DCL1, which in turn inhibit the PTGS process in plants and enhance the viral virulence effect [61]. In addition, the βC1 protein upregulates an endogenous RNAi suppressor calmodulin-like protein (CaM) and leads to the degradation of SGS3 and suppression of RDR6 activity, eventually affecting the antiviral RNA silencing [62,63]. V3 expressed during TYLCV infection, localizes in the Golgi apparatus, functions as an RNA silencing suppressor, and traffics along microfilaments to plasmodesmata to promote virus cell-to-cell movement [64,65]. Thus, it is common that geminiviruses encodes multiple proteins to suppress both TGS and PTGS by suppressing the activity or accumulation of RNA-silencing components. 

## 4. Protein-Kinase-Mediated Immunity

Protein kinases regulate the biological activity of many proteins by phosphorylation, and they play important roles in various plant biological processes, including defense [66]. Some protein kinases are reported to regulate plant defense against geminiviruses [10,67,68,69]. Sucrose non-fermenting1-related protein kinase 1 (SnRK1) is a Ser/Thr kinase, widely recognized as a key regulator of plant responses to various physiological processes, operating multi-organ crosstalk and potentially regulating downstream transcription factors to maintain cellular homeostasis [70]. SnRK1 belongs to the conserved kinase family and consists of a α catalytic subunit and β and γ regulatory subunits [71]. The overexpression of SnRK1 makes plant more resistant to geminivirus infection [72,73]. Geminiviral Rep interacts with Rep-interacting kinase (GRIK), an upstream activator of SnRK1, and their interaction stabilizes GRIK accumulation and activates SnRK1 to phosphorylate Rep [74,75,76]. SnRK1 interacts with βC1 encoded by TYLCCNB and CLCuMuB to reduce viral DNA accumulation and viral symptom severity by phosphorylating βC1. Phosphorylated βC1 fails to decrease DNA methylation and to upregulate rgs-CaM, thus impairing the suppression of both TGS and PTGS [68,77,78,79]. SnRK1 also phosphorylates the AL2/C2 protein to limit geminivirus infection [73]. Geminiviral C2 inactivates host SnRK1 and adenosine kinases through protein–protein interactions [48,80]. SnRK1 and ADK form a complex in plants, and alterations in either one may influence the others’ activity [81]. SnRK1 also inhibits translation by phosphorylating the cap-binding proteins eIF4E and eIFiso4E to condition antiviral defense. It is also inhibited by geminivirus pathogenicity factors [82]. These results suggest that SnRK1 interacts with and phosphorylates multiple viral proteins to control geminivirus infection.

Mitogen-activated protein kinases (MAPKs) play a crucial role in defense against diverse pathogens, including geminiviruses. MAPKs are activated during geminiviral infection and restrict geminiviral pathogenicity [83,84,85]. TYLCCNV infection activates MPK6/MPK3 and MPK4, although viral βC1 limits MAPK cascade-regulated defense by inhibiting MKK2 and MPK4 kinase activity [69]. Recently, TLCYnV C4 has been reported to interfere with MAPKs-mediated defense responses by inhibiting the dissociation of the ERECTA/BKI1 complex [86]. These findings illustrate the vital role of MAPK cascade in plant defense against geminiviruses. 

Receptor-like kinases (RLKs) regulate cell differentiation, development and innate immunity [87]. Several NSP-interacting RLKs (NIKs) interact with NSPs from distinct geminiviruses [88,89]. NIK confers a broad-spectrum tolerance to begomoviruses by suppressing viral translation [90,91]. Deficiency of NIK displays increased susceptibility to geminiviral infection [90,91]. However, NSP suppresses NIK activity to prevail over NIK-mediated resistance against geminivirus [88,92]. The TYLCV C4 protein interacts with many plant RLKs, including CLV1, FLS2, BRI1 and two plasma-membrane- and plasmodesmata-localized barely any meristem (BAM) 1 and 2 [93,94]. BSCTV C4 interacts with CLV1, which regulates the expression of an antiviral factor (WUSCHEL) [95]. In addition, C4 may suppress PTGS by interacting with BAM1/2 [96]. 

Several geminiviral genes, such as C4/AC4, are reported to interact with many Shaggy-like protein kinases [97]. Shaggy-like protein kinase SKη negatively regulates brassinosteroid (BR) signaling [98]. C4–SKη interactions are critical for C4 multifunctions, including viral symptom induction, RNA silencing suppression, cell cycle and BR signaling regulation, the induction of hyperplasia and cell division [99,100]. These findings demonstrate that there are different protein kinases pivotal in plant defense against geminiviruses, and geminiviruses exploit various strategies to suppress protein-kinase-mediated defense for effective infection.

## 5. Effector-Triggered Immunity (ETI)

Plant immune systems have evolved multilayer receptor systems to sense and induce pathogen defense responses. ETI restricts the pathogen at the site of infection (local resistance) by inducing programmed cell death (PCD), a phenomenon known as hypersensitive response (HR). Geminiviral proteins are both the inducers and suppressors of HR. Rep, C2 and V2 proteins are able to induce HR, meanwhile C4 and C2 are reported to antagonize HR [101,102,103,104]. These findings suggest that there exist natural antiviral R genes that confer resistance against geminiviruses. Indeed, *CYR1* encodes 1176 amino-acid-resistant proteins with a coiled structure at the N-terminus, central nucleotide-binding site (NBS) and C-terminal leucine-rich repeats (LRRs), conferring resistance against MYMIV by recognizing viral coat protein in *Vigna mungo*. Tomato Ty-2 also encodes a CC-NB-LRR R protein, which confers resistance against TYLCV by recognizing the TYLCV Rep/C1 protein [105,106,107]. 

## 6. Autophagy-Mediated Antiviral Defense

Autophagy is an evolutionarily conserved cellular activity that plays important roles in plant–pathogen interactions. During incompatible plant–virus interactions, autophagy prevents cells from death beyond viral infection sites [108]. Autophagy also plays an antiviral role in geminivirus infection by degrading viral proteins. βC1 and C1 from geminiviruses interact with autophagy-related gene 8 (ATG8) proteins and are degraded by autophagy [109,110] (Figure 2). The disruption of autophagy by silencing either ATG5 or ATG7 enhances geminivirus infection, while enhanced autophagy by silencing autophagy negative regulator GAPCs reduces geminivirus infection [109]. Interestingly, βC1 from CLCuMuB induces autophagy by disrupting the interaction of GAPCs and ATG3 [111]. CLCuMuB βC1 is degraded by autophagy. CLCuMuB βC1-mediated autophagy may reduce viral virulence, enhance host cell survival and enable successful infection during plant–virus co-evolution [112]. Recently, TYLCCNB-encoded βC1 is reported to induce the expression of NBR1 and interact with NBR1 in the cytoplasm to form granules. These cytoplasmic granules can prevent the degradation of viral βC1 by NbRFP1-mediated UPS-dependent degradation, leading to an increased βC1 accumulation and many severe disease symptoms [111,113]. Apart from its antiviral defense, autophagy may also contribute to geminivirus infection. rgs-CaM promotes TYLCCNV infection by interacting with suppressor of gene silencing 3 (SGS3) to mediate its autophagic degradation [114]. In this study, TYLCCNV infection is inhibited by the silencing of Beclin1, PI3K or VPS15, suggesting that autophagy may be required for TYLCCNV infection [114]. Furthermore, UVRAG and ATG14 (subunits of PI3K complex) are reported to contribute to geminivirus infection [115]. The effect of silencing Beclin1, PI3K, VPS15 or ATG14 on geminiviruses may depend on some other PI3P-dependent, non-autophagic membrane trafficking activity [115]. 

## 7. Ubiquitin-Proteasome System (UPS)-Mediated Anti-Geminiviral Defense

Ubiquitination is a post-translational modification process that is a major protein-degradation mechanism in plants. Three enzymes, namely the ubiquitin-activating enzyme (E1), the ubiquitin-conjugating enzyme (E2), and E3 ubiquitin ligase (E3) are required for ubiquitination [116]. Several studies have suggested a correlation between ubiquitination and geminivirus infection [85,117,118,119,120]. Silencing of either UBA1 (ubiquitin-activating enzyme) or RHF2a (RING-type E3 ubiquitin ligase) enhances TYLCSV infection [121,122]. The BSCTV C4 protein induces RKP, a RING finger E3 ligase, and affects geminivirus infection by regulating plant cell cycle [119]. Tobacco RFP1 interacts with TYLCCNB βC1 and prompts βC1 degradation via the ubiquitin-mediated 26S proteasomal pathway to attenuate viral symptoms [123]. In addition, CLCuMuB βC1 protein can disrupt the integrity of the SKP1/Cullin1 (CUL1)/F-box (SCF) complex SCF^COI1^ by interacting with s-phase kinase-associated protein 1 (SKP1), thereby disrupting plant ubiquitination and promoting viral infection and symptom induction [124]. UBC3 (ubiquitin-conjugating enzyme 3) activity is also blocked by βC1 [118]. The C2 proteins of TYLCSV, TYLCV and BCTV are reported to impair the derubylation of SCF E3 ligase complexes and inhibit jasmonate signaling by interacting with CSN5 [117,125]. CLCuMuB βC1 could enhance CLCuMuV accumulation, at least partially by repressing JA responses by interfering with plant ubiquitination [124].

## 8. Hormone-Mediated Defense against Geminivirus

Plant hormones are small, structurally unrelated molecules that not only regulate plant growth and development, but are also essential in plant defense against viral pathogens [126,127]. Several studies have highlighted the involvement of various phytohormones, such as salicylic acid (SA), jasmonic acid (JA), ethylene, auxin, cytokinin, gibberellic acid, brassinosteroids and abscisic acid, in plant–geminivirus infection [10,128]. The use of exogenous SA and JA improves resistance to TYLCV infection in plants [129]. SA, ethylene and cytokinin pathways genes are upregulated within geminivirus infections [85,130,131,132,133]. Whereas, the genes in JA and auxin pathways are differentially regulated in geminivirus infections [130,134,135,136]. Geminiviral C2 interacts with CSN5 and alters the derubylation activity of the CSN complex, which affects downstream signaling pathways, such as those of auxin, gibberellic acid (GA), ethylene (ET), salicylic acid (SA) and JA [117]. The C2 protein of geminivirus has also been shown to downregulate the expression of certain defense genes in the JA-mediated signaling pathway [137]. Geminiviral βC1 suppress JA-mediated defense by repressing JA downstream markers or by interacting with MYC2 andAS1 [125,138]. Furthermore, the ßC1 protein encoded by TYLCCB suppresses JA-dependent plant terpene biosynthesis to subvert plant resistance [138]. Geminiviral C4 interacts with auxin biosynthetic enzymes and disrupts endogenous auxin content [139]. The relationship between plant hormone pathways and geminiviruses has previously been well reviewed [128,140].

## 9. Conclusions

The *Geminiviridae* family is one of the largest families of DNA viruses infecting numerous crops and weeds (dicots and monocots). It also causes severe yield losses worldwide. Plants pose multilayered and comprehensive antiviral strategies to manipulate virus, such as RNA silencing, plant signaling, hormone signaling, protein degradation and so on. To make the microenvironment suitable for geminivirus infection, geminiviruses encode various proteins to interfere with host antiviral mechanisms, including the manipulation of the cell cycle, DNA replication, intra- and inter-cellular movement and the suppression of gene silencing and other antiviral defenses, such as the response to defense-related hormones. Viruses also usurp host-protein-degradation processes in order to reduce host defense, reduce cell death and promote viral replication. Geminiviruses co-evolve in long term plant–virus infection, and defense and counter-defense mechanisms in plant–geminivirus interactions are perplexing. Recently, a CRISPR/Cas9 system has emerged as a great tool to integrate geminivirus resistance [141,142]. Cas9-mediated immunity in tobacco enhanced resistance to cotton leaf curl disease (CLCuD) and African cassava mosaic virus (ACMV) [143,144]. In addition, the CRISPR/Cas9 system enhances resistance to TYLCV in tomato [145]. Plants could also possess other defense pathways against geminivirus, in addition to the defense pathways described above [146,147,148,149,150]. For examples, plants CMD1, CMD2 and CMD3 confer phenotypic disease tolerance to geminivirus with unknown mechanisms [151,152,153,154]. The Ty-5 gene encodes the mRNA surveillance factor Pelota, and its loss-of-function allele impairs viral translation, leading to viral tolerance, indicating that the Pelota gene is a susceptibility gene for multiple geminiviruses, including TYLCV [155,156]. In addition, plants recognize Ca^2+^ flux triggered by injuries to plant cells as the common molecular pattern of different viral infections to prime antiviral RNAi defense [38]. Recently, Yang et al. (2021) found that vacuolar acidification is required for plant antiviral defense against a positive-strand RNA virus–barley stripe mosaic virus (BSMV). Meanwhile, BSMV replicase γa inhibits the acidification of vacuolar lumen and suppresses autophagic degradation to promote viral infection by interacting with the V-ATPase catalytic subunit [157]. Many plant RNA viruses have evolved to suppress or manipulate host autophagy to promote viral infection [158,159]. Whether geminiviruses suppress or manipulate autophagy and how its underlying mechanisms work need to be of further concern. The identification of new host factors involved in virus infection that interact directly or indirectly with virus-encoded proteins is essential for the establishment of novel antiviral strategies.

## Figures and Tables

**Figure 1 viruses-15-00510-f001:**
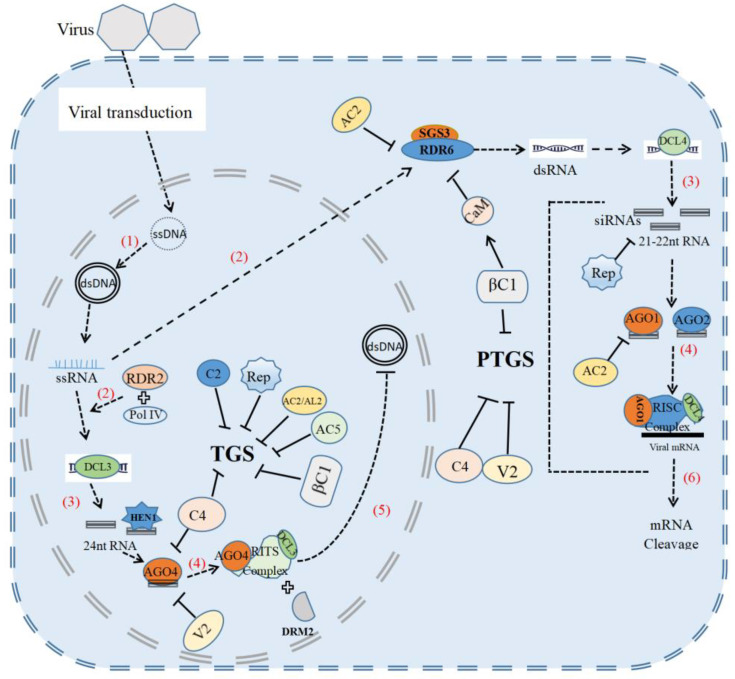
Antiviral RNA silencing and viral suppressors. (1) In the nucleus, viral ssDNA is converted to dsDNA. (2) The virus then uses host RDR2 or RDR6 to convert ssRNA to dsRNA. (3) dsRNA is processed into 21-22 or 24 nt siRNAs mediated by Dicer (DCL) where, in the nucleus, the 24 nt siRNAs are stabilized by HEN1. (4) Argonaute 4 (AGO4) and AGO1/2 interact with siRNA to form the RNA-induced transcriptional silencing complex (RITS) and RNA-induced silencing complex (RISC), respectively. (5) In the nucleus, RITS targets the viral transcribed genome and then interacts with structural domain-rearrangement methyltransferase 2 (DRM2) to achieve transcriptional gene silencing (TGS) of the viral genome. (6) In the cytoplasm, RISC mediates post-transcriptional gene silencing (PTGS), which inhibits the transcription of viral genes by degrading viral mRNA. In order to successfully infect, viruses produce several viral suppressors (VSRs). For example, in the nucleus, βC1, AC2, C2, C4, Rep, AC5 can interact with key components of the TGS pathway to help the virus resist transcriptional gene silencing. C4 and V2 promote viral infection by interacting with AGO4. In the cytoplasm, the viral βC1, C4, Rep and V2 proteins can similarly have a role in inhibiting post-transcriptional gene silencing and promoting viral infection. In addition, the viral AC2 protein can interact with host AGO1 and RDR6 to inhibit RNA silencing, and rgsCaM can inhibit post-transcriptional gene silencing by inhibiting the binding of RDR6 to SGS3. In addition, the βC1 protein of the virus can stimulate the accumulation of the rgsCaM protein to a certain extent.

**Figure 2 viruses-15-00510-f002:**
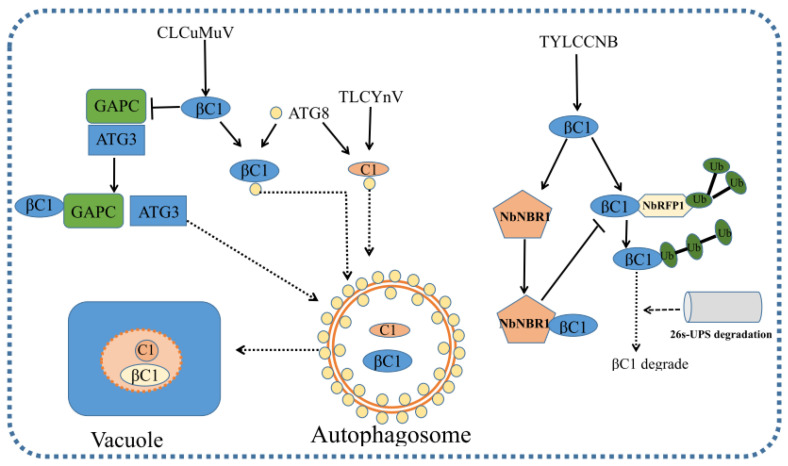
Autophagy in plant–geminivirus infection. CLCuMuB βC1 and TLCYnV C1 interacts with autophagy-related gene 8 (ATG8) protein and are degraded by autophagy. CLCuMuB βC1 bound to GAPCs and disrupted the interaction between GAPCs and autophagy-related protein 3 (ATG3) to induce autophagy. The βC1 of TYLCCNB can be degraded by the ubiquitin 26S proteasome system (UPS) mediated by NbRFP1 in *N. benthamiana*. In order to successfully infect the host, the viral βC1 protein induces the overexpression of NbNBR1 in the host, and then βC1 forms particles with NbNBR1 in the cytoplasm, which prevent βC1 from being degraded by the UPS system, resulting in an increased accumulation of the βC1 protein in the host cells and many severe symptoms.

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
