# Peer review of "Plant Defense and Viral Counter-Defense during Plant–Geminivirus Interactions"

_viruses, 2023, doi:10.3390/v15020510_

Round 1
Reviewer 1 Report
The manuscript is well written providing information on plant-virus interaction and crosstalk. Authors have provided adequate review on the topic along with comprehensive figures.
English editing is required (some sentences/words are highlighted to indicate).
Moreover, while addressing the strategies for developing viral resistance based on understanding of host-virus interaction, some strategies may be cited for providing examples such as (CRISPR/dCas9-mediated inhibition of replication of begomoviruses)
Author Response
Thank you for your comments. We have added CRISPR /Cas9 system mediated resistance strategies against geminivirus on our revised manuscript, see line 351-355.

Reviewer 2 Report
The manuscript viruses-2192456 “Plant Defense and Viral Counter-defense during Plant-Geminivirus Interactions” is a very interesting and complete review of the mechanisms of defense of plants against geminiviruses and the counter-defense of these pathogens to achieve infection. The paper summarizes, in a clear and understandable way, a large part of the knowledge that is currently available in the field, allowing to have a complete overview of the topic. I think the manuscript should be published after some comments are included. In addition, the conclusions are a very good summary of what has been developed, but the authors, as experts in the subject, should indicate where the investigations should be directed and why they believe that these knowledges are important in the field of plant pathology.

Author Response
Thank you for your positive feedback and comments. We have followed your suggestion and revised our manuscript. Please see revised manuscript (line 361-367) with Track Changes.
